# Pattern transition recognition based on transfer learning for exoskeleton across different terrains



Yifan Gao[1], Jianbin Zheng[1], Yang Gao[1], Ziyao Chen[1], Jing Tang[2] and Liping Huang[3]

[1] School of Information Engineering, Wuhan University of Technology, Wuhan, Hubei, China
[2] School of Electrical and Electronic Engineering, Hubei University of Technology, Wuhan, Hubei, China
[3] School of Computer and Information Engineering, Henan University of Economics and Law, Zhengzhou, Henan, China

## ABSTRACT

Human motion intention detection is a growing trend in wearable robots. In the study, a novel transfer learning method based on temporal convolutional network spatial attention (TCN-SA) is applied for pattern transition recognition under triple physical loads on different terrains. The proposed approach is used to recognize eight locomotion modes transition among five dynamic locomotion modes in sequence, such as level ground walking, stair ascending, stair descending, ramp ascending, and ramp descending. To address the problem of pattern transition recognition, transfer learning adapts a model from the source domain to the target domain. Temporal convolutional network (TCN) relies on local relationships in time sequence and gains steady gradient propagation. Furthermore, spatial attention (SA) provides insight into significant components in multi-dimensional feature selection. Pattern transition recognition based on a transfer learning method achieves higher accuracy and earlier prediction time (Pre-T). The accuracy of pattern transition detection reaches 97.46%, 97.62%, and 98.21% in M0, M20, and M40, respectively. In the process of pattern transition recognition, Pre-T of next locomotion mode in M0, M20, and M40 are 240–600 ms, 200–410 ms, and 120–420 ms before the step into that locomotion mode. The proportion of prediction time in a gait cycle (Pre-T/GC) in M0, M20, and M40 is 14.2–36%, 14.82–28.22%, and 7.4–29.1%, respectively. Ultimately, the results indicate that the proposed approach fulfills the expected performance in Pre-T and comparisons with TCN-attention, TCN, residual network (ResNet), and long short-term memory (LSTM) in assessment criteria. Our study early detects pattern transition, allowing the exoskeleton to traverse between adjacent terrains smoothly.

## INTRODUCTION

Over the last several decades, wearable devices have been widely developed for various applications. Exoskeleton robot is regarded as a potential technology for assisting human movement in recent research. Several potential research directions are discussed to support

Corresponding author
Jianbin Zheng,
papsub2020@gmail.com

the development of the comprehensive and efficient benchmarking methodology for exoskeleton robot (*Pinto-Fernandez et al., 2020*). The latest key technologies and research topics are analyzed and summarized, including mechanical structure, human-machine interaction, adaptive control strategies, and evaluation methods for power-assisted walking efficiency (*Qiu et al., 2022*). Human motion intention recognition is essential to realize compliant control in lower limb exoskeletons. Precise and real-time recognition of human motion intention promotes adaptive and reliable human-machine interaction. Human motion intention recognition is crucial in advancing lower limb exoskeleton technology (*Li et al., 2023*). Pattern transition recognition is an important component of human motion intention recognition in lower limb exoskeleton. At present, extensive research has been conducted on advancing suitable locomotion mode recognition methods for lower limb exoskeleton. These methods are aimed at identifying both steady-state locomotion modes and transient transitions in real time, which is essential for realizing seamless transition between adjacent terrains. It facilitates human–machine cooperation and ensures user safety.

Multiple types of wearable sensors are applied for recognizing many locomotion modes and patterns transition in the situation of human walking. *Parri et al. (2017)* put forward the real-time pattern recognition method of human activities in lower extremity exoskeleton. The mixed classifier consisted of the time-based approach based on gait kinematics and the fuzzy-logic method using encoders and force sensitive resistors (FSRs). *Martinez-Hernandez & Dehghani-Sanij (2018)* presented the adaptive Bayesian inference system to recognize three locomotion modes and gait events using inertial measurement units (IMUs). *Tang et al. (2024)* studied the dense convolutional network (DenseNet) and long short-term memory (LSTM) with a channel attention mechanism (SENet) method (SE-DenseNet-LSTM) for pattern detection using IMUs on three terrains. *Sahoo et al. (2020)* developed the prototype for early prediction of locomotion mode transition using range sensors and FSRs. *Tiwari & Joshi (2020)* designed the inexpensive wireless gait event detection approach using infra-red range sensors.

Muscle synergy analysis (MSA) is the descending dimension method that decomposes abundant muscle excitations into slight temporal synergy excitations. MSA is applied to pathological population, which is aimed at better understanding the intrinsic physiological characteristics reflected in muscle activity. *Liu & Gutierrez-Farewik (2021)* presented the muscle synergy identification method based on long short-term memory (LSTM) to predict knee joint moments using surface electromyogram signal (sEMG). *Ao et al. (2020)* studied that PCA estimated unmeasured muscle excitations through synergy excitations extracted from muscles using sEMG. *Song, Ma & Liu (2023)* proposed online joints angles prediction method based on LSTM using sEMG. *Huo et al. (2018)* came up with fast locomotion mode detection method based on the body sensor system of IMUs and sEMG. Valid human motion intention prediction thoroughly understood how the brain prepared for body movement. *Bai et al. (2011)* put forward the computational method to predict human movement before it occurred from electroencephalograph (EEG).

On diverse terrains, it is crucial to detect pattern transition between neighbor states. *Joshi & Hahn (2016)* designed the classification framework of transition type to detect

direction (ascent/descent) and terrain (ramp/stair) with sensors fusion of sEMG and accelerometry. *Hagio, Fukuda & Kouzaki (2015)* evaluated the motor control during gait transition based on muscle synergy. It is essential to detect human movement intention for compliant control of exoskeleton. *Godiyal et al. (2018)* came up with gait phase detection method based on force myography in steady state and transition between overground and ramp. *Saito et al. (2018)* put forward the lower limb muscle synergy identification method using sEMG during treadmill running on level and inclined ground. *Angelidou & Artemiadis (2023)* proposed the subject-specific locomotion mode recognition approach using sEMG and kinematic to detect human intent to transition in the compliant surface. *Mundt et al. (2020)* developed the neural network models to predict ground reaction force and joint moments based on joint angles in gait analysis using the 3D motion capture system.

In locomotion-related activities, it is significant to recognize transition between standing and sitting. *Rattanaphon et al. (2020)* studied the continuous EEG rhythms decoding approach in the stage of action observation, motor imagery, and motor execution for standing and sitting. They assigned the motional work of transition between standing and sitting in performance. *Liu et al. (2019)* put forward the three-step control approach to detect transition between siting and standing using two IMUs in knee exoskeleton. *Chen et al. (2024)* proposed the bidirectional long short-term memory (BiLSTM), attention mechanism, and convolutional neural network (CNN) method (CNN-BiLSTM-Attention) to segment and identify transition between siting and standing in inertial sensors. *Wang et al. (2022)* proposed the multi-feature fusion method in process of motion conversion from squat/sit to stand. It contributed to recognize pattern transition in rehabilitation training.

Currently, deep learning methods are widely applied to locomotion modes transition in assisted exoskeleton. *Young & Hargrove (2016)* studied user-independent intent recognition systems to perform seamless transition between adjacent states. Fall prevention and detection was critical on research of elderly healthcare and humanoid robot. *Jain & Semwal (2022)* proposed the preimpact fall detection system based on deep learning, which contributed to relieve injuring from falls. *Qian et al. (2022)* studied the locomotion mode recognition method based on depth sensor and the gait phase estimation approach based on adaptive oscillator for terrain-adaptive assistive walking. *Varol, Sup & Goldfarb (2010)* developed the control strategy and intent detection method to identify 90 patterns transition in the real-time powered prosthesis.

However, few studies focus on pattern transition recognition in outdoor activity. This article put forward a transfer learning approach based on temporal convolutional network spatial attention (TCN-SA) for pattern transition recognition under triple physical loads on different terrains. In the open air, there are five locomotion modes, such as level ground walking (LW), stair ascending (SA), stair descending (SD), ramp ascending (RA), ramp descending (RD). There are eight patterns transition between adjacent locomotion modes, such as LW→SA, LW→SD, LW→RA, LW→RD, SA→LW, SD→LW, RA→LW, RD→LW. To address similar target problems, transfer learning adapts the model from source domain to target domain. Transfer learning demonstrates better in detecting

pattern transition on diverse terrains. Temporal convolutional network can feasibly capture local dependency in time sequence and obtain stable gradient propagation. What's more, spatial attention is adapted to identify and focus on crucial information in multidimensional features selection. Eventually, multigroup experiments indicate that the transfer learning method based on TCN-SA reaches the expected effect of pattern transition recognition in Pre-T and Pre-T/GC. It makes outstanding influence in comparisons with usual machine learning methods in assessment criteria.

In view of the importance of pattern transition recognition in outdoor activity, the prominent innovations of our study are mentioned below.

- A novel transfer learning model based on TCN-SA is designed for the sake of pattern transition recognition in multilevel loads on diverse terrains.
- In transfer learning model, TCN can availably acquire local dependencies in temporal series by stacking convolution layers and extending the receptive field of convolution kernel. Moreover, spatial attention enhances model sensitivity to highlight the important specific spatial region of multiple features.
- It extracts multidimensional features of joint angle to the transfer learning model and then fine-tunes hyperparameters to recognize pattern transition, which achieves the predicted effect in the average accuracy.
- In the process of pattern transition recognition, Pre-T of next locomotion mode in three physical loads are 240–600 ms, 200–410 ms, and 120–420 ms before the step into that locomotion mode. Pre-T/GC in M0, M20, and M40 are 14.2–36%, 14.82–28.22%, and 7.4–29.1%, respectively.

The main content of this study is arranged in turn. "Materials" depicts experimental protocol, the equipment configuration, data preprocessing, dataset architecture and evaluation indicator. "Methods" explains transfer learning, TCN, attention, spatial attention, and framework of transfer learning model based on TCN-SA in detail. "Results" displays the hyperparameters comparison, performance of transfer learning model based on TCN-SA, Pre-T in pattern transition, and Pre-T/GC in pattern transition. "Discussion" describes performance assessment and comparison to the methods. Finally, "Conclusions" makes conclusion and expresses limitation and future work.

## RELATED WORK

The general control architecture of the powered hip and knee exoskeleton is made up of three control levels. In high-level control, it pays attention to human motion intention detection. In mid-level control, it determines desired assistive trajectories, joint angles and joint moment. In low-level control, it directly regulates the actuators to track the desired assistive trajectories and then adjusts the self-state of exoskeleton. Human motion intention detection is significant for exoskeleton to supply precise control. Pattern transition recognition is an important part of human motion intention detection. This study focuses on mainly pattern transition recognition module based on transfer learning to realize seamlessly transition between neighbor locomotion modes.

So far, deep learning approaches are generally employed to detect locomotion modes transition in assistive devices. *Bruinsma & Carloni (2021)* designed different deep neural network models for real-time prediction of eight dynamic locomotion modes and 24 locomotion modes transition among them in one transfemoral osseointegration amputee using IMUs. *Liu, Wang & Huang (2016)* proposed the portable terrain recognition module using the laser distance meter and the IMU. It predicted terrain transition before the step requested to convert the prosthesis control mode. To develop assistive devices, *Grimmer et al. (2020)* estimated the biomechanics of the transition between stair ambulation and level ground walking. *Peng et al. (2016)* studied that the individuals adjusted their biomechanics in anticipation of walking-stair transition. *Cheng et al. (2022)* put forward the control framework of modeling the joint kinematics on their steady-state and transitional gaits between adjustable incline and stairs in powered prosthesis. *Soo & Donelan (2010)* presented that step-to-step transitions were separated from mechanical walking data of other participants for the purpose of assigning advance and return movements. Moreover, the ankle joints contributed to the task of redirecting the velocity of the mass center during forward swing.

Pattern prediction was applied to the seamless switching of exoskeleton controllers to support human walking on various terrains. *Li et al. (2024)* constructed the multimodal frame of deep belief network to detect pattern and forecast pattern transition assignments. It is fuzzy to walking stability assessment in gait transitional phases (loading and unloading) due to the method limitation and mixed-use biomechanical signals. *Mahmood, Raza & Dehghani-Sanij (2022)* put forward that Nyquist and Bode methods utilizing neuromechanical output/input responded to estimate the stability of gait transition. *Stolyarov, Burnett & Herr (2018)* proposed that the real-time movement tracking method was developed to estimate knee and ankle joint translations using inertial signals in powered prosthesis across three terrains.

## MATERIALS

### Experimental protocol

This study involved human participants. All ethical and experimental procedures were approved by the ethical committee for sports science experiments in Beijing Sport University (Application No. 2019007H) and conducted in accordance with the protocols for human motion experiment.

Fifteen healthy participants (5F/10 M, Height: 173.2 ± 1.7 cm, Weight: 69.8 ± 4.8 kg, Age: 28.2 ± 1.3 years) are voluntary and supplied written informed consent to take part in the study. As shown in Figs. 1A–1E, the participants with exoskeleton engage in walking activities in scene. All the participants were divided into three groups. They walked on three terrains of level ground, stair, and ramp, severally. Each group held up multilevel loads (0, 20, 40 kg) on the back of human body respectively. M0, M20, and M40 denote multilevel loads of 0, 20, and 40 kg, severally. First group went straight ahead along the concrete road. Second group climbed up the stairs and then went back the same way. Third group went up the ramp with a slope gradient and then followed it back to level ground. Finally, three groups got through the walking process described above.

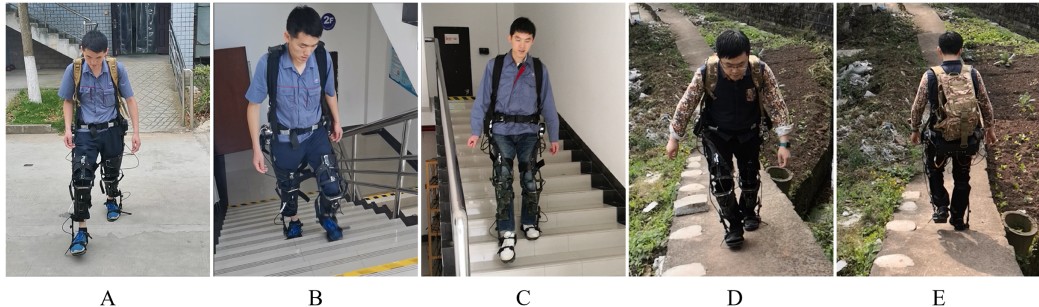

**Figure 1** The subjects wearing exoskeleton participate in walking in scene. (A) LW. (B) SA. (C) SD. (D) RA. (E) RD.                

## The equipment configuration

The STM32F765II microcontroller applies the advanced ARM Cortex-M7 32-bit to the development board in the system. Data transmission goes through RS485 bus and operates at the speed of 2 Mb/s with the sampling frequency of 100 Hz. PC equipped with AMD Ryzen 7 6800H CPU, 32 GB RAM, and Windows 11 (64-bit) serves as the data processing unit. It is Anaconda 3, Python 3.10, and PyTorch 2.0 in the software environment. A random algorithm is executed to enable real-time performance verification using multithreading.

## Data preprocessing

Each participant wears the lower limb exoskeleton. FSRs and IMUs attached to feet, shanks, thighs and trunk of both mechanical legs in exoskeleton are applied to acquire spatial and temporal data. The body trunk along vertical direction and the thigh are computed to gain hip angle. The thigh and the shank are calculated to gain knee angle. The foot IMU in horizontal direction serves as the foot angle. Clinical gait analysis of human body is applied to define joint angles.

It is indispensable to preprocess original data to promote multidimensional features consistent and achieve optimal performance. The original data is calculated into ground reaction force and joint angle, which apparently expresses the diversity in gait analysis. Degree is used as the unit for measuring joint angle. The calculated joint angles are used as inputs to the dataset. The packet loss takes advantage of linear interpolation to compensate the loss in data delivery. Furthermore, joint angle data conducted by normalization plays as the input data for model training. Therefore, it is significant to preprocess chain in the dataset.

## Evaluation indicator

Comprehensive evaluation index system composed of Pre-T, Pre-T/GC, confusion matrix, accuracy, precision, recall, and F1 is used to assess performance. Each of these indexes is interpreted as shown below.

Prediction time (Pre-T) is defined as the interval between a vital moment and the point of heel strike. A vital moment represents predicted point in the machine learning model. Pre-T/GC represents the proportion of a gait cycle duration consumed by prediction time.

$$\mathrm{Pre-T}/GC = \frac{t}{T} \tag{1}$$

where $t$ denotes prediction time, T signifies a gait cycle.

The confusion matrix is generally used to evaluate classification performance. It consists of four key components and summarizes the model prediction by comparing them with true labels. True positive (TP) indicates that positive samples are correctly predicted as positive. True negative (TN) denotes that negative samples are correctly predicted as negative. False positive (FP) refers to negative samples that are incorrectly predicted as positive. False negative (FN) refers to positive samples that are incorrectly predicted as negative.

Accuracy represents the ratio of correctly classified samples to the total number of samples.

$$\mathrm{Accuracy} = \frac{TP + TN}{TP + TN + FP + FN}. \tag{2}$$

Precision is the ratio of true positive predictions to the total number of samples predicted as positive.

$$\mathrm{Precision} = \frac{TP}{TP + FP}. \tag{3}$$

Recall is the ratio of true positive predictions to the total number of actual positive samples.

$$\mathrm{Recall} = \frac{TP}{TP + FN}. \tag{4}$$

F1-score combines precision and recall into a single metric by calculating their harmonic mean.

$$\frac{2}{F_1} = \frac{1}{\mathrm{Precision}} + \frac{1}{\mathrm{Recall}}. \tag{5}$$

It is essential to acquire the assessment criteria to generate differences in comparative approaches. Every experiment is repeatedly conducted to verify the statistical rule of the network model.

## METHODS

### Situation of pattern transition

There are multiple locomotion modes transition across diverse terrains, such as level ground, stair, and ramp. It is composed of standing phase and swing phase in a gait cycle depicted in Fig. 2. Heel strike not only acts as the end point of last gait cycle, but also plays as the starting point of subsequent gait cycle. Heel off regards as the dividing line of standing phase and swing phase. Pre-T is defined as the interval between a vital moment

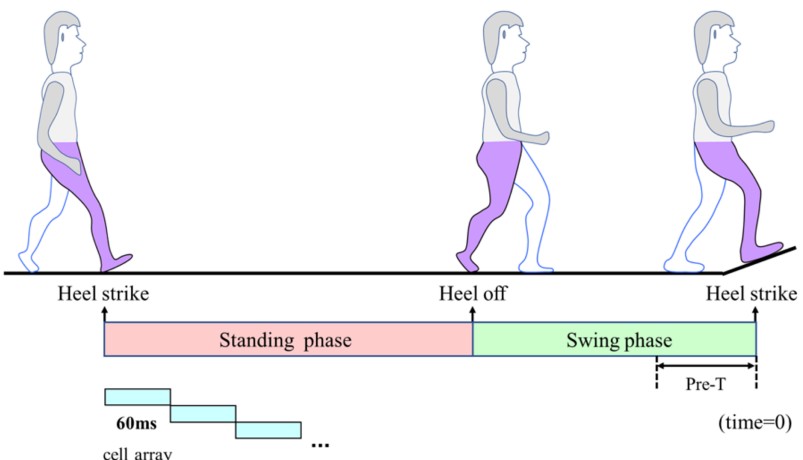

**Figure 2 The gait cycle consists of standing phase and swing phase.** Pattern transition occurs in swing phase.

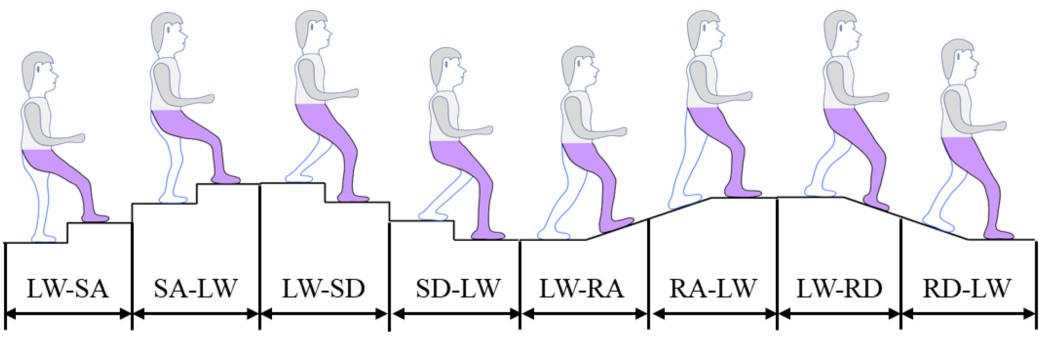

**Figure 3 Eight locomotion modes transition involved in the dynamic states.** The lower extremity is shown in purple.

and the point of heel strike. Pattern transition appears a critical interval time before subsequent locomotion mode. In addition, a series of cell array are located in preceding transition to subsequent locomotion mode. The length of cell array is 60 ms. As described in Fig. 3, Pattern transition is explicit between two locomotion modes adjacently, such as LW→SA, LW→SD, LW→RA, LW→RD, SA→LW, SD→LW, RA→LW, RD→LW.

## Dataset architecture

All the datasets consist of three types of sub-datasets in three levels of weight, which come from human walking on different terrains. Pattern transition occurs in swing phase of the gait cycle in the curves of hip angle, knee angle, foot angle, and foot pressure depicted in Figs. 4A–4H. Pattern transition is explicit between two locomotion modes adjacently, such as LW→SA, LW→SD, LW→RA, LW→RD, SA→LW, SD→LW, RA→LW, RD→LW. It is eight patterns transition among five dynamic locomotion modes in sequence, such as LW, SA, SD, RA, and RD. The proportion of the training set and the testing set in a dataset is 4:1. The input features are composed of left and right hip angle, left and right knee angle,

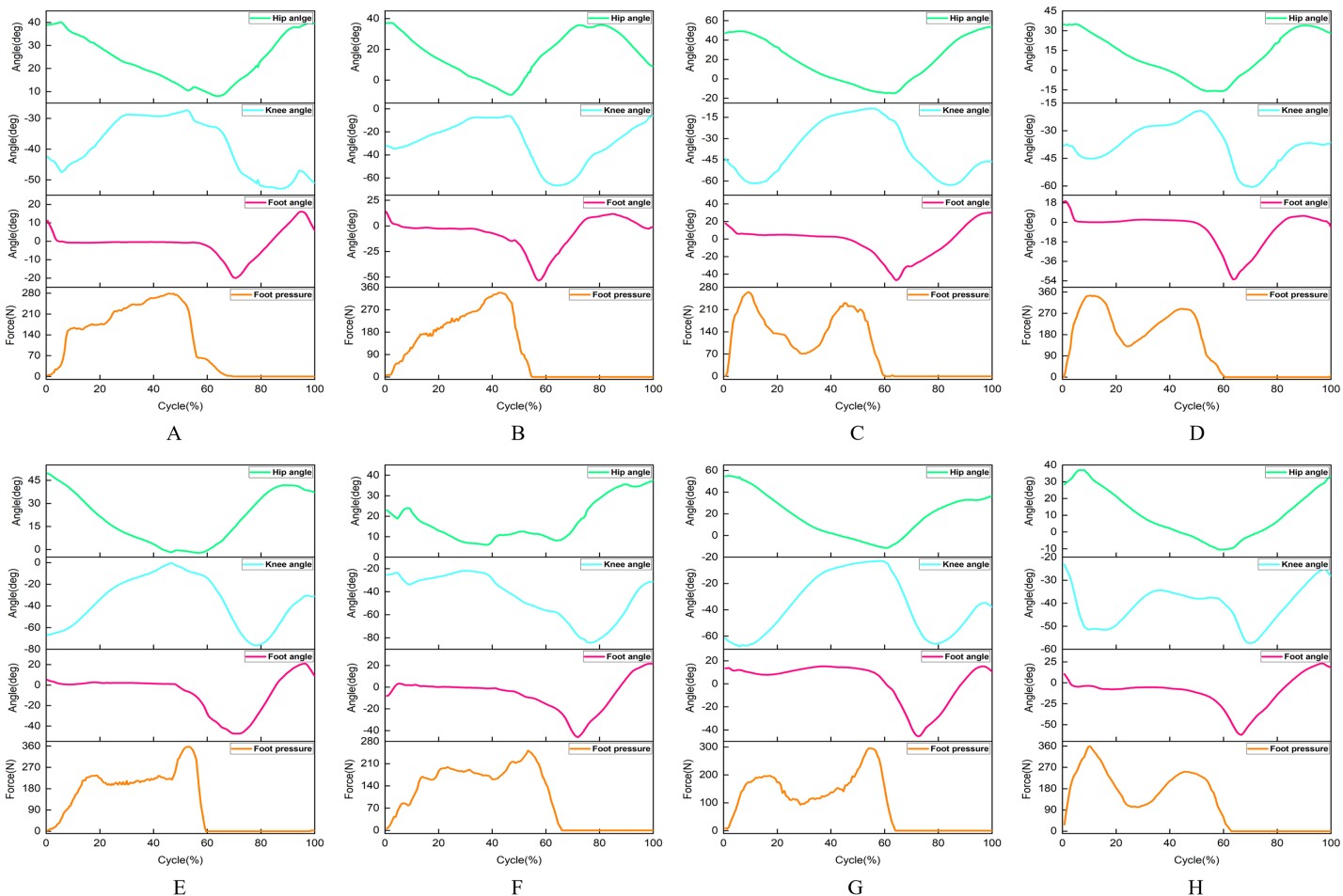

**Figure 4** (A–H) Gait cycle curves of hip angle, knee angle, foot angle, and foot pressure in every locomotion mode transition.

and left and right foot angle. The multiple features and the length of the unit sample are set to 6 and 6 in input layer, respectively. The matrix size of a cell array consisting of the unit sample data is set to 6 × 6.

## Transfer learning

Transfer learning attracts widespread attention in the rapid development of artificial intelligence. It is designed to transfer knowledge and adapts the model from source domain to target domain. It contributes to reduce the sample size requirement of target domain and accelerate learning process, which improves generalization performance of the model. There are primarily transfer learning methods, such as model transfer, feature transfer, and relationship transfer. As shown in Fig. 5, model transfer refers to migrating the model from source domain to target domain and fine-tuning hyperparameters to adapt to classification task in the target domain.

The model transfer method involves multiple steps, such as choosing a source model, determining a transfer strategy, adjusting model hyperparameters, and evaluating model

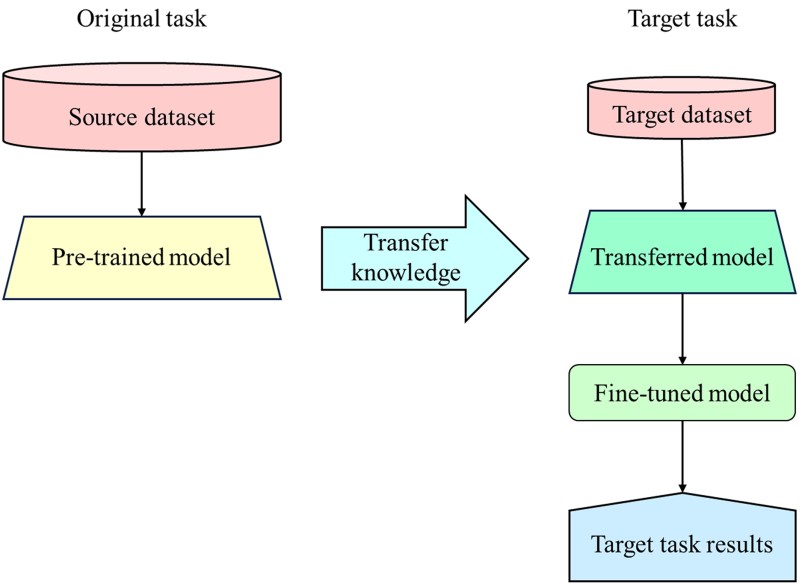

**Figure 5** **The conceptual model of transfer learning.**

performance. It is necessary to select a suitable pre-trained model to meet the demand of the target task and the characteristics of source domain. An appropriate transfer strategy depends on the higher similarity between the source domain and target domain. It fine-tunes the model hyperparameters on the target domain, which usually involves in adjusting the parameters of the preceding layers and the final layer to satisfy the needs of the target assignment. The model performance on the target domain is estimated by evaluation metrics. As an efficient machine learning method, transfer learning has advantage on addressing challenges of data scarcity and domain differences.

## TCN

As shown in Fig. 6, temporal convolutional network (TCN) is a model framework of deep learning developed for processing serial data. It combines the parallel processing capability of CNN with the long-term dependency of recurrent neural network (RNN) and makes it a powerful tool for sequence modeling tasks. It presents the core characteristics of sequence modeling and then defines the network structure.

An input sequence $x_0, x_1, \ldots, x_T$ and the corresponding output $y_0, y_1, \ldots, y_T$ are at each time step. When predicting the output $y_t$ is at a specific time step $t$, the input $x_0, x_1, \ldots, x_t$ are used to observed. A sequence modeling network is an arbitrary function $f : X^{T+1} \to Y^{T+1}$ that generates mapping as follows.

$$\hat{y}_0, \hat{y}_1, \ldots, \hat{y}_T = f(x_0, x_1, \ldots, x_t). \tag{6}$$

To satisfy the causality constraint, $y_t$ only depends on $x_0, x_1, \ldots, x_t$.

The learning objective in sequence modeling is used to search the network that minimizes the expected loss $L(y_0, y_1, \ldots, y_T, f(x_0, x_1, \ldots, x_T))$ between the actual outputs

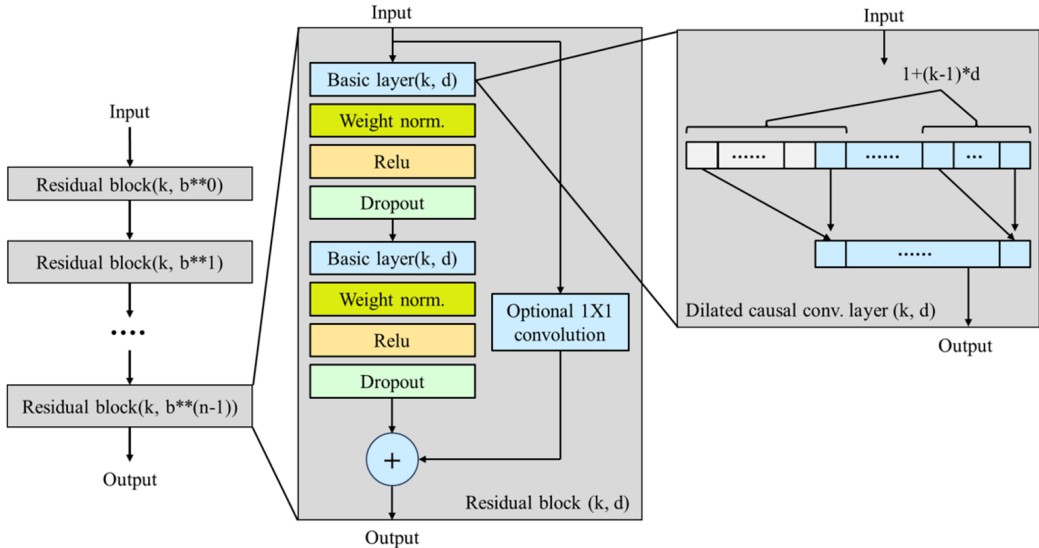

**Figure 6 The architecture diagram of TCN.**

and the predicted values, where the sequences and outputs are drawn from the certain probability distribution.

**Causal convolution.** It means that the output of TCN model depends on the current and past points. In standard convolution operations, each output value is based on its surrounding input values, such as current points and past points.

However, the weights in causal convolution are applied to the current and past input values and ensures the directionality of the information flow and preventing future information from leaking into the current output. Zeros are typically padded on the right side of the convolution kernel so that the current and past data is used to compute the output.

$$y(t) = \sum_{i=0}^{k-1} f(i) \cdot x(t-i) \tag{7}$$

where $k$ refers to the kernel size, $x$ denotes the input series, $f$ represents the convolution kernel.

**Dilated convolution.** It is employed by TCN to expend the receptive field. The gaps of dilated convolution are inserted between kernel elements and allows for broader context capture. Therefore, it is able to grasp the contextual information in the input data. The dilation factor determines the spacing between elements in the convolution kernel. The mathematical representation of dilated convolution as follows.

$$y(t) = \sum_{i=0}^{k-1} f(i) \cdot x(t-d \cdot i) \tag{8}$$

where $d$ is the dilation rate.

**Residual connection.** TCN employs residual connections to alleviate the vanishing gradient problem and improve the training efficiency of deeper networks and optimize model performance. Residual connections are a key component of residual network (ResNet). In residual connections, the output of a certain layer in the network directly added a few layers to another layer, which forms a skip connection.

Given an input $x$, the output is $F(x)$ after going through several layers. The final output is $x + F(x)$. It allows gradients to flow directly back to earlier layers during backpropagation. It reduces the issue of vanishing gradients and enables effective training of deeper architectures. The output of a residual block is expressed.

$$\text{output} = \text{activation}(\text{input} + F(\text{input}))  \tag{9}$$

where $F$ represents the combination of the convolution layer and the activation function.

The basic structure of TCN consists of multiple residual blocks. Each residual block includes dilated causal convolution layer, layer normalization, ReLU activation function, and dropout layer.

## Attention

Attention mechanism allows the model to dynamically adjust the attention weights in processing data depicted in Fig. 7. It highlights important data and ignores irrelevant details. It imitates human brain to selectively focus on concrete details selectively, which demonstrates exceptional performance in processing sequential and high-dimensional data.

The input to attention mechanism typically is composed of three parts. Query (Q) represents the processed data currently in the current time step. Key (K) denotes the features of the reference data used to match query. Value (V) represents the content associated with each key and contributes to the output of the attention mechanism.

The attention mechanism calculates the relevance between the query and the key and then performs the weighted average of the values. The formula is mentioned below.

$$\text{Attention}(Q, K, V) = \text{softmax}\left(\frac{QK^T}{\sqrt{d_k}}\right)V  \tag{10}$$

where $QK^\top$ calculates the dot product similarity between the query and the key, the scaling factor $\sqrt{d_k}$ prevents gradient instability from excessive dot product values. Softmax converts the similarity into weights as probability distribution.

## Spatial attention

Spatial attention regards as the adaptive mechanism of spatial region selection to enhance model performance and efficiency in deep learning. Generally, spatial attention mechanism achieves enhanced representation of effective spatial features and suppression of irrelevant spatial features by gaining attention weights for each spatial position.

As shown in Fig. 8, spatial attention mechanism is conducted by the following steps. Assume that the feature map has a size of $CHW$, $C$ is the number of channels. H and W are the height and width, respectively.

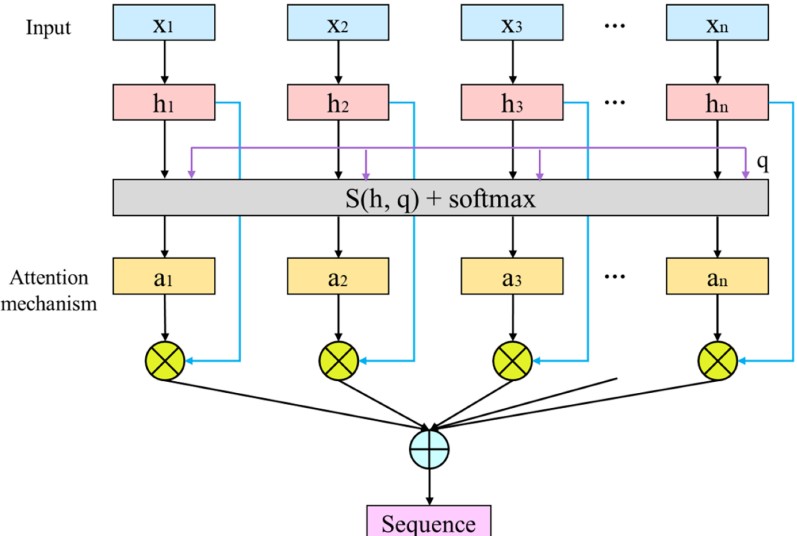

**Figure 7** **The structure of attention mechanism.**

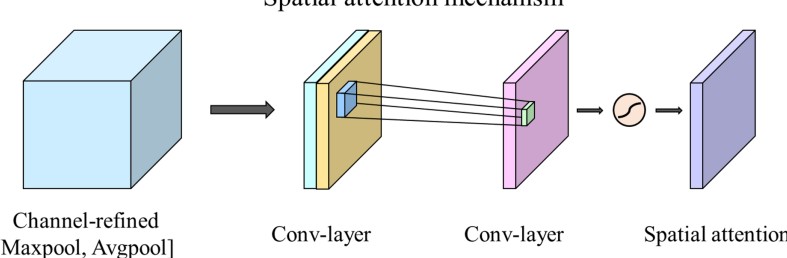

**Figure 8** **The structure frame of spatial attention model.**

**Feature mapping.** Each spatial position in the feature map is mapped to an attention score space. A small convolutional kernel implements the process. It is aimed at generating a corresponding attention weight for each spatial position.

**Weight calculation.** Attention weights for each position are computed and then normalized into a probability distribution by softmax. The sum of attention weights across all positions equals 1, which prompts the model to emphasize the significant areas and disregard less relevant ones.

**Weighted feature fusion.** Those weights are utilized to the original feature map that results in the weighted feature map. It enhances the feature representation of regions with high attention weights and suppresses those with low attention weights.

### Framework of transfer learning model based on TCN-SA

As shown in Fig. 9, it is the framework of transfer learning model based on TCN-SA. Transfer learning refers to migrate TCN-SA model from dynamic pattern detection to

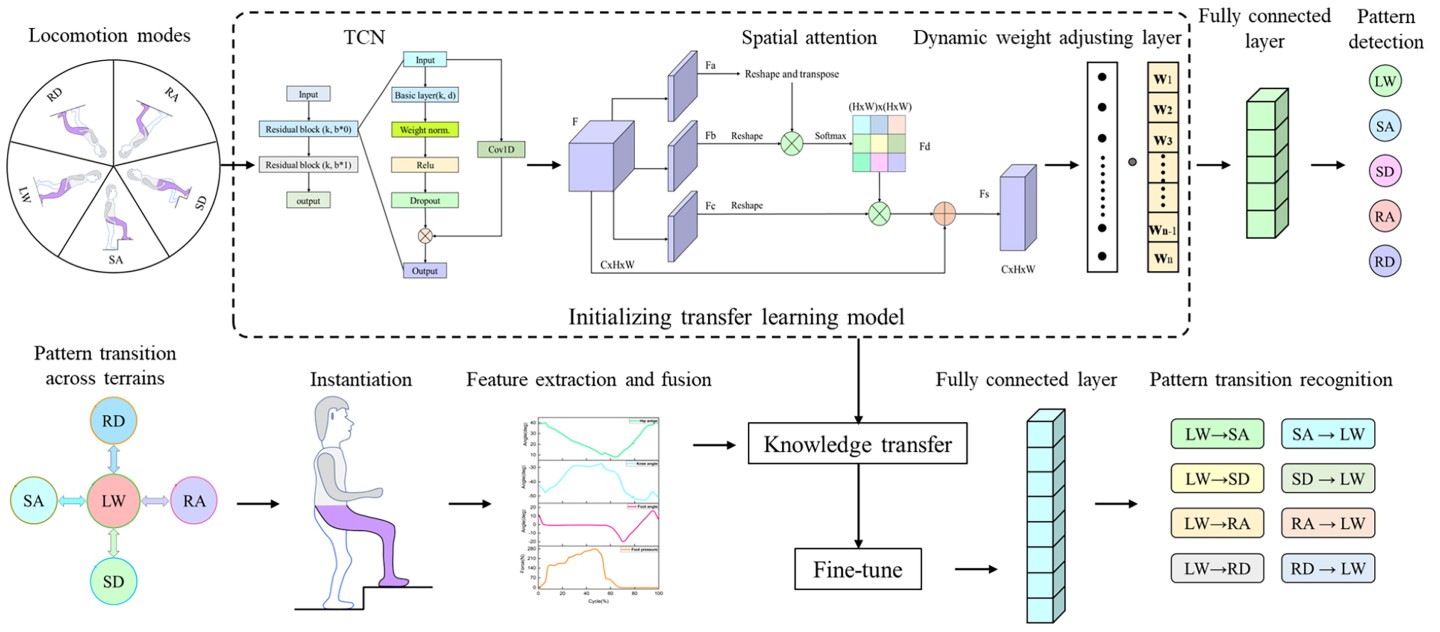

**Figure 9** The structure frame of transfer learning model based on TCN-SA.

classification task and fine-tunes hyperparameters to adapt to pattern transition recognition.

### Transfer learning model initialization

The exoskeleton wearers walk on various terrains of level ground, stair, and ramp, severally. Generally speaking, it is divided into five dynamic locomotion modes (LW, SA, SD, RA, RD). The joint angle data is collected by IMUs. The preprocessed joint angle data goes through TCN to acquire temporal sequence data. TCN are designed for sequence modeling and time-series analysis. It takes advantage of causal convolution that depends on past and current points to predict subsequent data at a given time step. It applies dilated convolutions to obtain long-term dependencies effectively. It exploits residual connections to raise the training efficiency of neural networks.

Spatial attention mechanism enhances neural network performance by focusing on most relevant spatial feature mapping. By weight calculation to assign attention weights to different spatial locations, it is necessary to prioritize critical features and suppress irrelevant and redundant information. It goes through weighted feature fusion to gain spatial relationships to improve the model performance, which makes it effective in the task of spatial dependence analysis.

The dense connectivity allows fully connected layer to extract complicated and high-level representations from the input data. By aggregating information across multidimensional features, fully connected layer is used for decision-making and feature combination and makes it effective in model performance. Therefore, the transfer learning model based on TCN-SA carries out pattern detection assignment.

*Knowledge transfer*

The exoskeleton wearer walks across neighboring terrains, which provides seamlessly transition between adjacent locomotion modes. The joint angle data goes through data preprocessing to get ready for features extraction and fusion. Multidimensional features data is put into the transfer learning model by knowledge transfer. Then, it is essential to fine-tune the hyperparameters of the transfer learning model. Moreover, it imports into fully connected layer and the transfer learning model training in order. At last, the transfer learning model based on TCN-SA recognizes pattern transition. Three sub-datasets in three physical loads execute the flow path in turn mentioned above.

## RESULTS

### The hyperparameters comparison

In terms of many locomotion modes transition across different terrains, the hyperparameters comparison experiments are mainly conductive to analyze the influence of channel numbers, kernel size, and learning rate on accuracy of pattern transition recognition shown in Fig. 10. Multigroup experiments in transfer learning model based on TCN-SA are conducted as follows.

Part 1: In the hybrid model, kernel size 6 and learning rate 0.005 are set as the constants. It performs the comparative experiments of the channel numbers. It demonstrates the relationship of channel numbers and accuracy in the proposed model. To verify the impact of channel numbers on accuracy, they are set to 24, 32, 48, 64, and 72 respectively in Fig. 10A. When the channel number is 48, the mean value reaches peak.

Part 2: In the hybrid model, the channel number 48 and learning rate 0.005 play as the constants. It shows the comparative experiments of kernel size. To confirm the impact of kernel size on accuracy of the hybrid model, they are set to 4, 6, 8, 10, and 12, severally in Fig. 10B. When the kernel size is 6, the mean value achieves a high level.

Part 3: In the hybrid model, the channel number 48 and kernel size 6 serve as the constants. Learning rate is applied to the comparative experiments. It displays the correlation of learning rate and accuracy in the proposed model. To verify the impact of learning rate on accuracy in the proposed model, they are set to 0.001, 0.005, 0.01, 0.015, and 0.02 respectively in Fig. 10C. When learning rate is 0.005, the mean value achieves the higher level.

### Performance of transfer learning model based on TCN-SA

In Table 1, the framework of the transfer learning model is evident in the aspect of the hyperparameters choice. The kernel size and the number of channels in the convolutional layer are 6 and 48 separately. In model training, the batch size and the epochs are set to 6 and 100, severally. The learning rate and the scale factor are 0.005 and 0.2, respectively.

As described in Figs. 11A, 11B, the performance of TCN-SA takes advantage of the layered K-fold cross validation. The K values are set to 3 and 5, severally. Each fold's score is in the range of 97% to 99%. In the training process, the dataset is divided into K groups with same size. K−1 folds serve as training and the Kth fold remains validation. In cross validation, the stability of the evaluation result largely depends on the K value. The layered
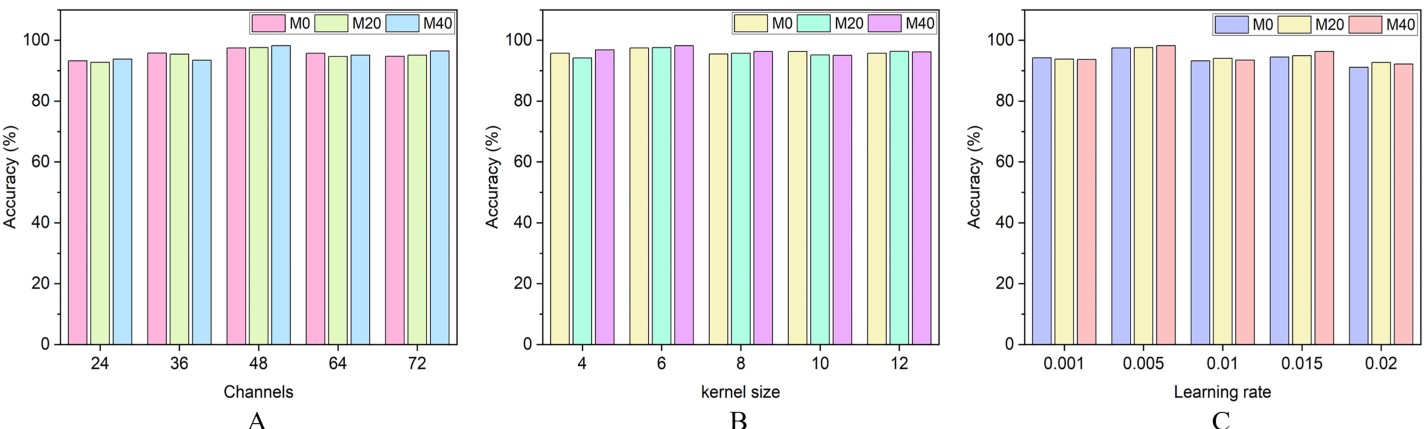

**Figure 10** (A–C) The channels, kernel size, and learning rate in the hyperparameters optimization.

**Table 1** The structure and parameters of the transfer learning model based on TCN-SA.

| Stages | Parameters | Values |
|---|---|---|
| Data preprocessing | Cell array | 6 |
| | Input size | 6 |
| Model structure | Kernel size | 6 |
| | Channels | 48 |
| | Learning rate | 0.005 |
| | Scale factor | 0.2 |
| Training parameters | Optimizer | Adam |
| | Batch size | 6 |
| | Epochs | 100 |

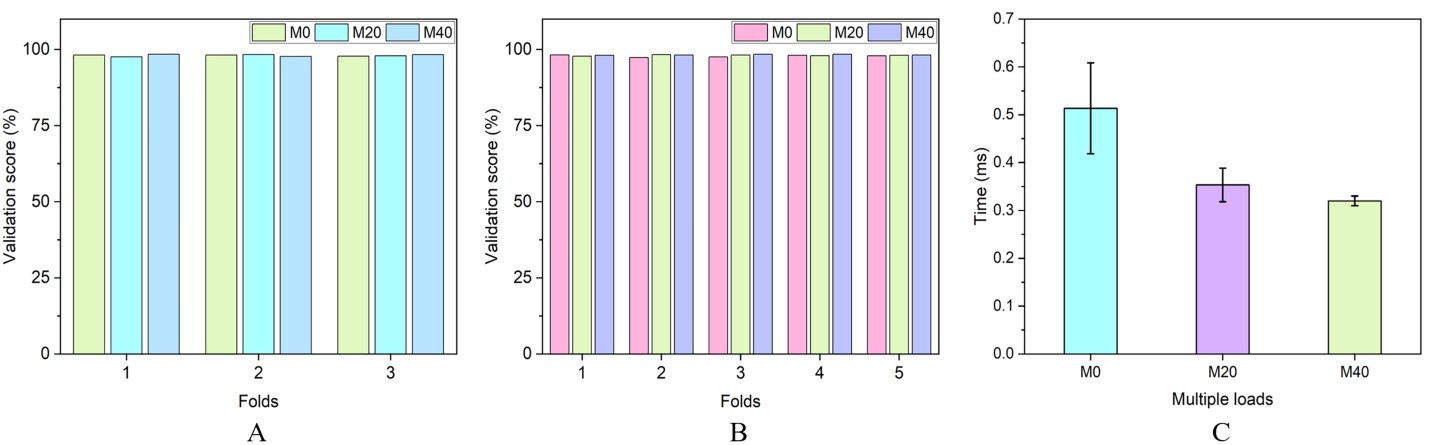

**Figure 11** (A, B) The K-fold cross validation of the transfer learning model based on TCN-SA in M0, M20, M40. (C) The time of the hybrid model in M0, M20, M40.

K-fold confirms that the sample proportion for each locomotion mode transition is steady in every fold.

It takes the calculation time in resource consumption to model training and testing once. As shown in Fig. 11C, the calculation time for multigrade loads is approximately 0.51 ms, 0.35 ms, and 0.32 ms, respectively. It reduces computational complexity and saves resource cost. To verify the prominent performance of TCN-SA on pattern transition recognition, the following experiments are carried out in time analysis.

## Pre-T analysis in pattern transition

Prediction time (Pre-T) is analyzed to verify the efficiency of the proposed model in the real-time process of pattern transition. The recognition effects of Pre-T in multilevel loads are depicted in Fig. 12. During eight locomotion modes transition, the hybrid model can display the real-time performance of detecting pattern transition between two neighboring states steadily.

Pre-T and the errors in M0 serve as the referenced time. In M20, Pre-T of LW→RD, RA→LW, LW→RA, SD→LW, SA→LW, and LW→SA declines rapidly. Pre-T of RD→LW and LW→SD grow gradually. The errors of LW→SD, SA→LW, and LW→SA rise rapidly. The errors of RD→LW, LW→RD, and RA→LW drop off dramatically.

During pattern transition recognition, Pre-T of next locomotion mode in three levels of weight are 240–600 ms, 200–410 ms, and 120–420 ms before the step into that locomotion mode. On the whole, the mean prediction time in M20 is more stable than that in M0.

In M40, Pre-T of LW→RD, RA→LW, LW→RA, SD→LW, LW→SD, SA→LW and LW→SA go down dramatically. Pre-T of RD→LW rises distinctly. The errors of LW→RD, RA→LW, LW→RA, SD→LW, and LW→SD drop off slowly. The error of LW→SA rises rapidly.

The average prediction time decreases distinctly in multigrade loads. The errors in three levels of weight are 65–220 ms, 45–170 ms, and 36–130 ms before Pre-T of next locomotion mode. In the proposed model, most locomotion modes transition between two adjacent states are steadily predicted before next locomotion mode.

## Pre-T/GC analysis in pattern transition

Pre-T/GC is shortened form of the ratio of prediction time to a gait cycle. Pre-T/GC is statistical to confirm the real-time performance of the proposed model during pattern transition. It is essential for identifying pattern transition between two states adjacently. Most transitions begin early in swing phase prior to next locomotion mode.

As described in Fig. 13, it emphasizes Pre-T/GC and the errors during eight locomotion modes transition in multigrade loads. In M0, their proportion of Pre-T/GC are more than 25% in LW→SA, SD→LW, and LW→RD, respectively. In M20, their ratio of Pre-T/GC are more than 25% in RA→LW and RD→LW. In M40, their ratio of Pre-T/GC are more than 25% in LW→SA, SD→LW, LW→RD, and RD→LW. For locomotion modes transition, Pre-T/GC in multilevel loads are 14.2–36%, 14.82–28.22%, and 7.4–29.1% in a gait cycle. The errors of Pre-T/GC in multigrade loads are 3.68–14.97%, 2.85–8.8%, and

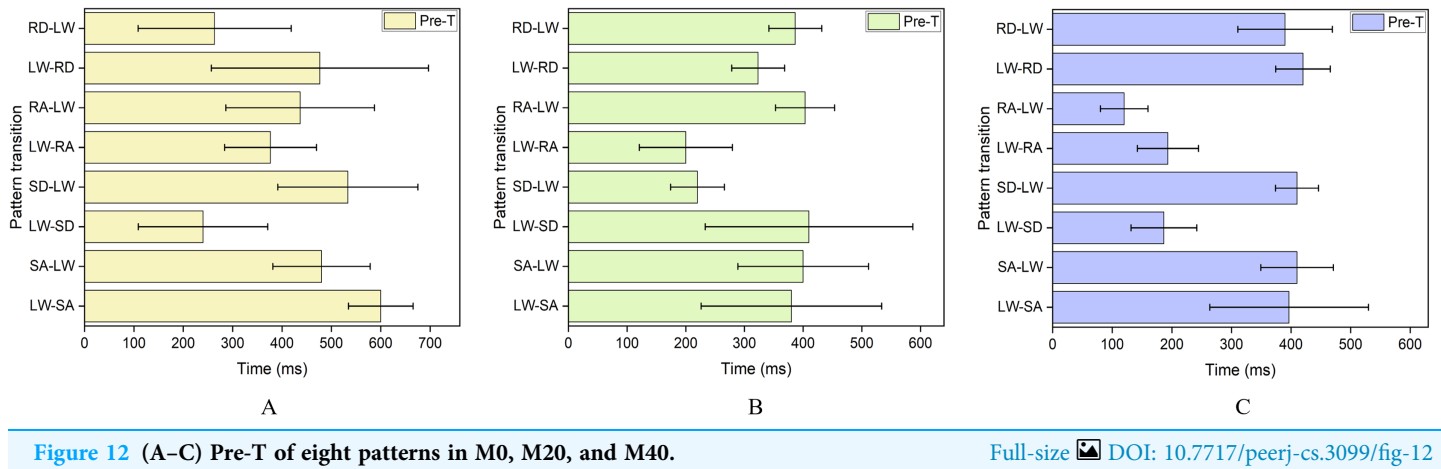

**Figure 12** (A–C) Pre-T of eight patterns in M0, M20, and M40.

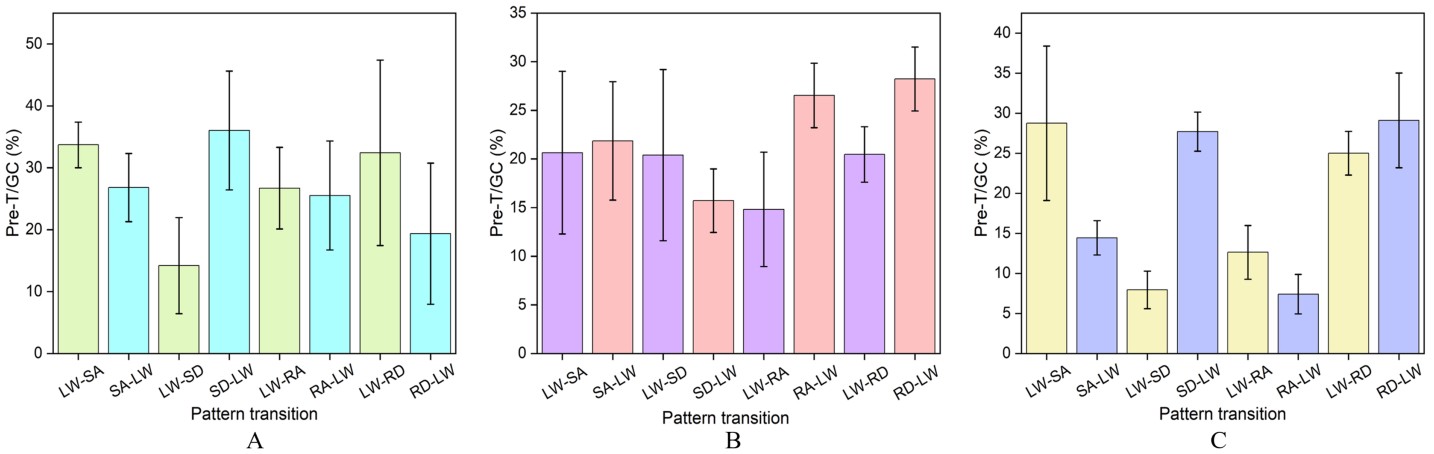

**Figure 13** (A–C) **The proportion distribution in eight patterns transition.** The ratio between prediction time and gait cycle in M0, M20, and M40.

2.14–9.65%, respectively. The proposed model can seamlessly detect transition between two states adjacently, which presents the real-time effect in triple physical loads.

## DISCUSSION

### Performance assessment

In this part, the transfer learning method based on TCN-SA has prominent performance on pattern transition detection by contrast with TCN-attention, TCN, ResNet, and LSTM in the assessment criteria of accuracy, precision, recall, and F1. Confusion matrix displays true values and predicted values in standard of division to grasp how the accuracy continuously changes.

In TCN-SA, Figs. 14A–14C depicts that the precision, recall, and F1 of eight locomotion modes transition in M0, M20, and M40 gradually go up. As described in Figs. 14D–14F, the accuracy of their confusion matrixes in multilevel loads achieves 97.46%, 97.62%, and 98.21%, severally.

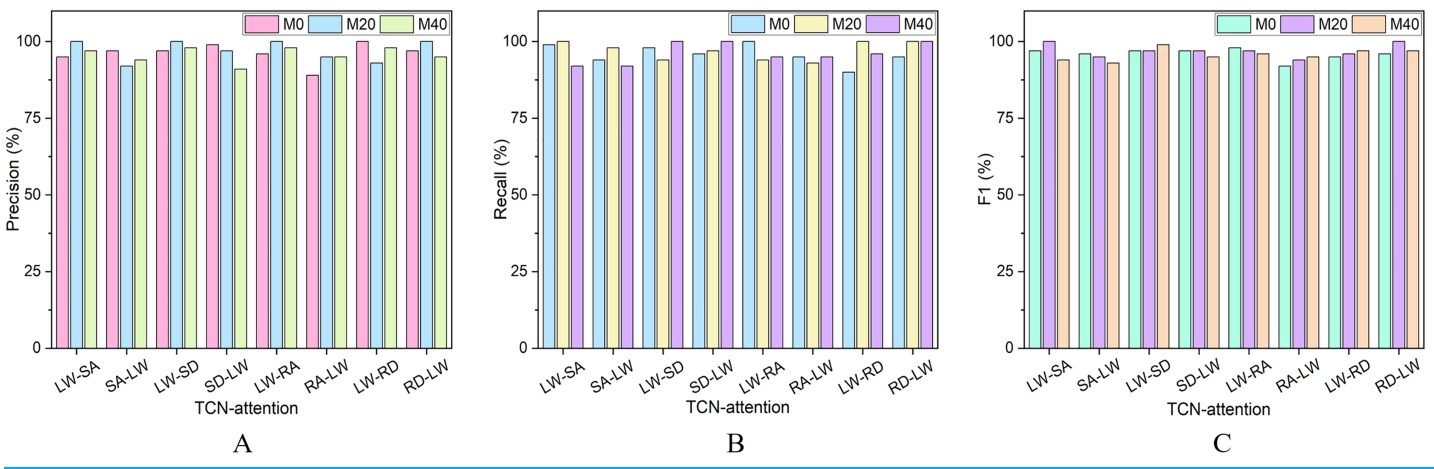

**Figure 14** (A–C) The performance evaluation of the transfer learning model based on TCN-SA in precision, recall, and F1. (D–F) The confusion matrixes of them for pattern transition recognition.

**Figure 15** (A–C) The performance evaluation of the TCN-attention model in precision, recall, and F1.

In TCN-attention, the accuracy of their confusion matrixes in multigrade loads reaches 96.38%, 96.43%, and 96.13%, severally. As shown in Figs. 15A–15C, the precision, recall, and F1 of eight locomotion modes transition in three levels of weight gradually increase and then go down.

In TCN, the accuracy of their confusion matrixes in M0, M20, and M40 achieves 95.89%, 96.83%, and 94.94%, severally. As displayed in Figs. 16A–16C, the precision, recall, and F1 of eight locomotion modes transition in triple physical loads gradually go up and then sharply decrease.

In ResNet, the accuracy of their confusion matrixes in M0, M20, and M40 reaches 96.35%, 95.12%, and 96.3%, respectively. As described in Figs. 17A–17C, the precision, recall, and F1 of eight locomotion modes transition in multilevel loads go down in difference.

In LSTM, the accuracy of their confusion matrixes in M0, M20, and M40 achieves 81.49%, 86.2%, and 88.52% severally. As illustrated in Figs. 18A–18C, the precision, recall, and F1 of eight locomotion modes transition in multigrade loads dramatically decrease.

Hence, the experimental results demonstrate that the accuracy of TCN-attention, TCN, ResNet, and LSTM is less than 97%. The transfer learning method based on TCN-SA has prominent effect in assessment criteria of precision, recall, and F1. It is apparent that the proposed method adequately improves the efficiency of pattern transition recognition.

## Generalization capability

As described in Fig. 19A, the comparative experiments of one-to-one and many-to-many groups are carried out to clarify the generalization capability. In the one-to-one experiments, lower limb motion data from a single subject is used for both training and testing the hybrid model. In the many-to-many experiments, lower limb motion data from multiple subjects is used to train the hybrid model, which is then evaluated on the preprocessed data from the separated multi-subject sample. The results of multiple experimental groups demonstrate that the hybrid model provides good performance for individual independence. The proposed method shows excellent consistency in recognizing pattern transition. Moreover, the hybrid approach performs strong generalization capability across 15 subjects in the self-constructed dataset.

As shown in Fig. 19B, it demonstrates the effectiveness of pattern transition recognition across many locomotion modes on two public datasets. In this study, two public datasets (GIOT-2024 and EPIC-2023) from EPIC Lab in Georgia Institute of Technology are the human lower-limb biomechanics and exoskeleton wearable sensors datasets. There are cyclic activities (level ground, stair ascending, stair descending, ramp ascending, and ramp descending) from several subjects without loads in normal speed in three terrains. The collected data from IMU and force plate attached to exoskeleton is at the sampling frequency of 200 Hz. The diversity of indoor activities provides a valuable basis for validating the robustness and generalizability of the proposed approach.

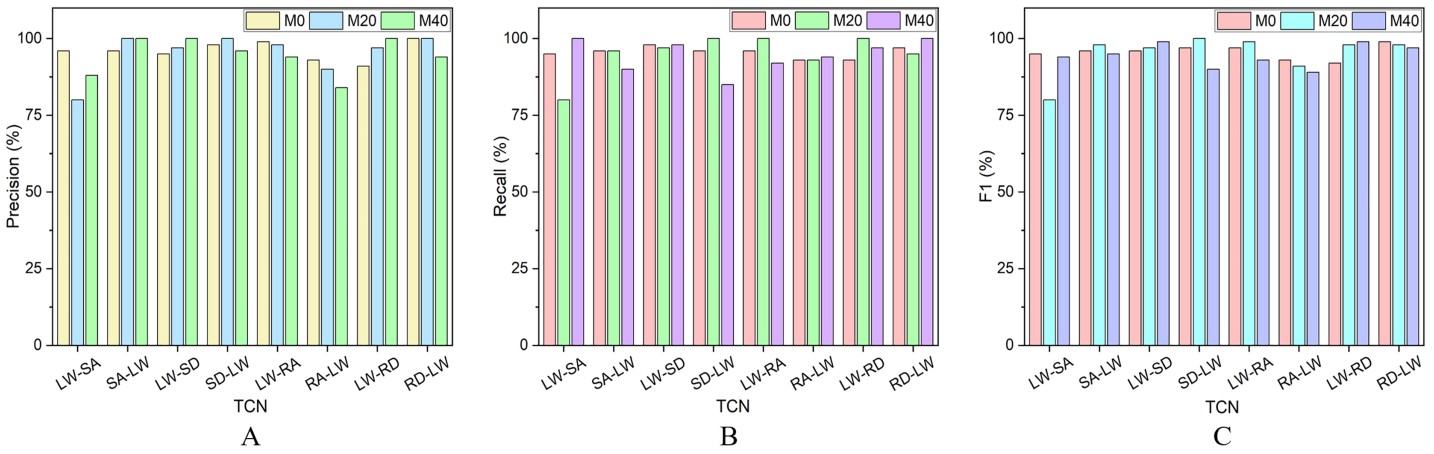

**Figure 16** (A–C) The performance evaluation of the TCN model in precision, recall, and F1.

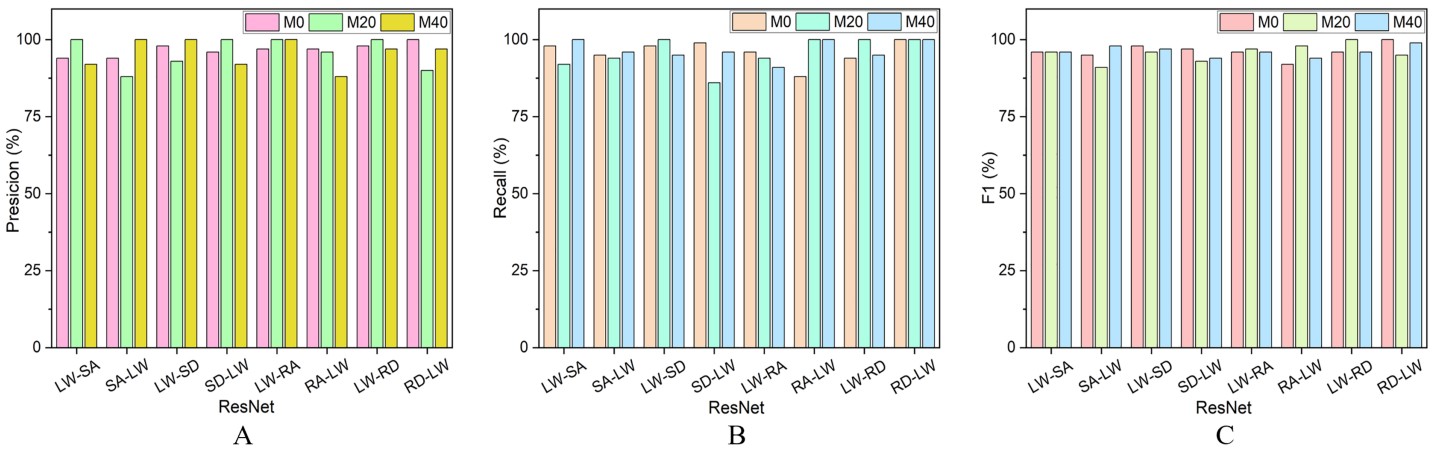

**Figure 17** (A–C) The performance evaluation of the ResNet model in precision, recall, and F1.

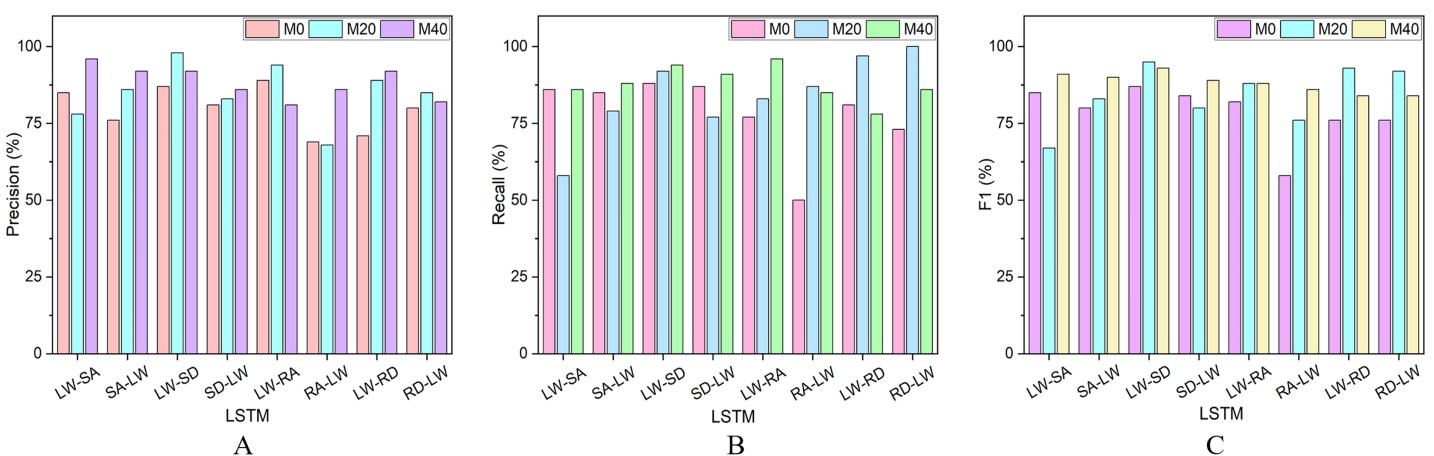

**Figure 18** (A–C) The performance evaluation of the LSTM model in precision, recall, and F1.

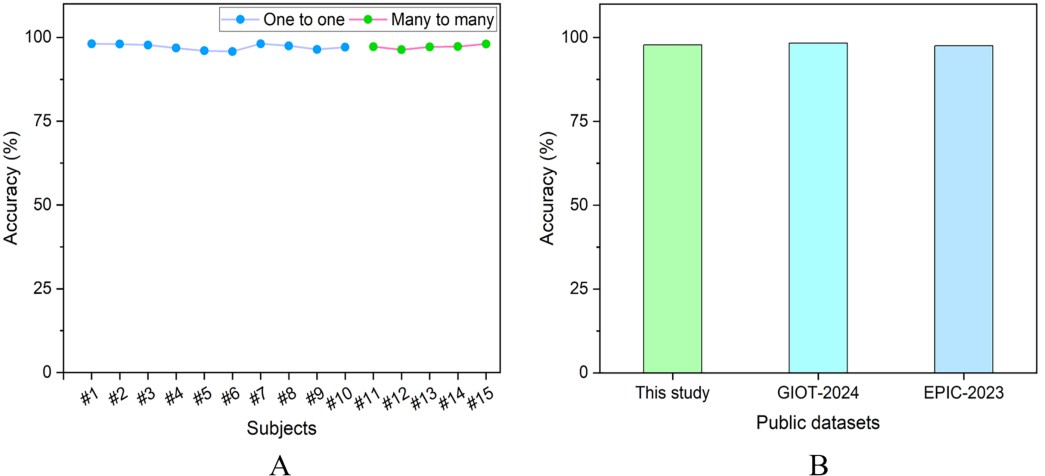

**Figure 19** (A) The relation of the accuracy and many subjects in pattern transition recognition. (B) The impact of pattern transition recognition on two public datasets.

**Table 2 Comparison to recent methods.**

| References | Methods | Scenes | Sensor type | Transition | Subjects | Accuracy (%) |
|---|---|---|---|---|---|---|
| *Figueiredo et al. (2020)* | Gaussian SVM | Outdoor | IMU | 9 | 10 Healthy | 97.65 |
| *Papapicco et al. (2021)* | Binary tree | Indoor | IMU | 4 | 10 Healthy | 95.6 |
| *Camargo et al. (2021)* | LDA, SVM, DBN | Indoor | EMG, IMU | 8 | 15 Healthy | 96 |
| *Su et al. (2019)* | CNN | Outdoor | IMU | 8 | 10 Healthy | 94.15 |
| | | | | | 1 Transtibial amputee | 89.23 |
| *Liu, Wang & Gutierrez-Farewik (2021)* | MSI | Indoor | EMG, IMU | 27 | 8 Healthy | 94.5 |
| *Liu & Gutierrez-Farewik (2023)* | MSP | Indoor | EMG, force plate | 7 | 9 Healthy | / |
| This study | TL-TCN-SA | Outdoor | IMU | 8 | 15 Healthy | 97.76 |

## Comparison to the approaches

In Table 2, this study is in contrast with many international advanced approaches in certain diversity background. We proposed the transfer learning method based on TCN-SA for eight locomotion modes transition in multilevel loads using IMUs on different terrains. Furthermore, the average accuracy of pattern transition recognition is 97.76%.

*Figueiredo et al. (2020)* studied that Gaussian support vector machine (SVM) model had the desired effect in nine locomotion modes transition recognition using IMUs. *Papapicco et al. (2021)* proposed that binary tree method predicted motion intention in next step using IMUs. *Camargo et al. (2021)* put forward the combined classifier of latent Dirichlet allocation (LDA), SVM, and deep belief network (DBN) for pattern transition detection using sEMG and IMUs. *Su et al. (2019)* studied CNN model for pattern

transition detection using IMUs in healthy and disabled participators. *Liu, Wang & Gutierrez-Farewik (2021)* came up with the muscle synergy-inspired (MSI) method of non-negative matrix factorization (NMF) and non-negative least squares (NNLS) for 27 locomotion modes transition using sEMG and IMUs. During pattern transition detection, prediction time of next step was about 300–500 ms before the step into that locomotion mode. *Liu & Gutierrez-Farewik (2023)* demonstrated that muscle synergy patterns (MSP) method based on NMF, NNLS and LSTM for seven locomotion modes transition using sEMG and force plate reached accurate moment prediction.

Hence, it is evident that internal factors of wearable sensors significantly affect pattern transition detection, such as sensor types and corresponding application scopes. There is a clear distinction between indoor and outdoor scenes in current studies. In addition, the differences between healthy and mobility-impaired individuals reported in many studies may lead to variation in detection performance.

## CONCLUSIONS

We put forward the transfer learning method based on TCN-SA for pattern transition recognition in multilevel loads across diverse terrains. Collecting human movement data from 15 subjects using IMUs for transfer learning model training and validation. The experiments of hyperparameters choice are conducted to improve the transfer learning model framework. Moreover, the experiments of eight locomotion modes transition in multigrade loads are carried out to verify the prominent performances of Pre-T and Pre-T/GC. In contrast with TCN-attention, TCN, ResNet, and LSTM, the proposed method has excellent effect on pattern transition detection. Experiment results demonstrate the strength of the proposed method in assessment criteria of accuracy, precision, recall, and F1. The proposed method makes good difference in pattern transition recognition under triple physical loads that paves the way for research and development in the future.

## LIMITATION AND FUTURE WORK

Up to now, our approach has good effect on capability of pattern transition recognition on diverse terrains, such as flat ground, stair, and slope. The population group of this study is limited to healthy adult. Human motion data is acquired by the restricted biomechanical sensors attached to wearable robot, such as IMUs and FSRs. In the near future, pattern transition detection is expected to be conducted in rugged terrains, such as swamps and hillsides. The dynamics and kinematics of human walking may vary across different ethnic groups from various regions, such as Asia, Europe, Africa, and Latin America. A 3D camera will be integrated with wearable robot to recognize complex terrains in the field.

## ACKNOWLEDGEMENTS

The authors deeply appreciate Jianbin Zheng, Wei Zhou, Jue Lu, Yu Wang of the Intelligent Sensing and Control Research Group for providing the experimental conditions.

### Funding

This research was funded by the National Key Research and Development Program of China (NO. 2017YFB1300502) and Henan Provincial Science and Technology Research Project (NO. 252102220122). The funders had no role in study design, data collection and analysis, decision to publish, or preparation of the manuscript.

### Grant Disclosures

The following grant information was disclosed by the authors:
National Key Research and Development Program of China: 2017YFB1300502.
Henan Provincial Science and Technology Research Project: 252102220122.

### Competing Interests

The authors declare that they have no competing interests.

### Author Contributions

- Yifan Gao conceived and designed the experiments, performed the experiments, analyzed the data, performed the computation work, prepared figures and/or tables, authored or reviewed drafts of the article, and approved the final draft.
- Jianbin Zheng conceived and designed the experiments, authored or reviewed drafts of the article, and approved the final draft.
- Yang Gao analyzed the data, prepared figures and/or tables, and approved the final draft.
- Ziyao Chen performed the experiments, performed the computation work, prepared figures and/or tables, and approved the final draft.
- Jing Tang analyzed the data, authored or reviewed drafts of the article, and approved the final draft.
- Liping Huang performed the experiments, authored or reviewed drafts of the article, and approved the final draft.

### Data Availability

    The datasets and software code are available in the Supplemental Files.

### Supplemental Information

Supplemental information for this article can be found online at http://dx.doi.org/10.7717/peerj-cs.3099#supplemental-information.

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
