# Peer review of "Pattern transition recognition based on transfer learning for exoskeleton across different terrains"

_PeerJ Computer Science, doi:10.7717/peerj-cs.3099_

## Round 0.1 · original submission · Major Revisions

Reviewer 1 ·

Basic reporting

This manuscript proposes transfer learning method based on TCN-SA in pattern transition recognition under triple physical load on diverse terrains. The related simulation and experimental verifications are provided to illustrate the effectiveness of the proposed active and passive control scheme of exoskeleton. Hence, the manuscript is interesting and the presentation is well-written.

The author can refer to the following comments, and improve the contribution and presentation in the revision.
(1) What is the different of the proposed exoskeleton from the other exoskeleton, especially in active and passive control scheme?

(2) How to guarantee the wearable comfort performance of the proposed exoskeleton to different users?

(3) Since many users have different motion styles and features, the designed exoskeleton should consider more information about individual feature.

(4) How about the stiffness of the mechanical structure? While, the exoskeleton mass is still considered for the wearable exoskeleton since the users cannot bear too heavy exoskeleton mass.

(5) Some new references also discussed gait identification and control problem similar to the author’s exoskeleton, such as multilevel control strategy of human-exoskeleton cooperative motion, and gait planning and multimodal human-exoskeleton cooperative control based on central pattern generator. The authors may give more descriptions about related work.

(6) There exist many typos and grammar errors in the manuscript. Please carefully check all the presentation in the revision.

Experimental design

no comment

Validity of the findings

no comment

Additional comments

no comment

Reviewer 2 ·

Basic reporting

The manuscript proposes a pattern transition recognition based on the transfer learning method, which achieves the prominent effect of higher accuracy and earlier prediction time for 12 locomotion modes transition among 7 dynamic patterns. The manuscript is relatively straightforward with sufficient literature survey. However, the article did not adequately compare methods with the latest studies, and problems with unclear descriptions, inconsistencies, and English readability needed to be improved. Comments are given to help the authors improve this paper’s quality before reconsideration.

1. The introduction section should not be a single paragraph.

Experimental design

2. How many subjects are involved in the study? Table II lists 10 subjects were participated in the experiments while Experimental Protocol (Page 3) lists 15 subjects.
3. “M40, M20, and M0 denote multilevel loads of 40 kg, 20 kg, and 0 kg, severally.” is written in the “The hyperparameters comparison” section, which should be moved to the protocol or experiment setup section.
4. Why are different load conditions explored in the experiment for the gait transition recognition task, but not speed/cadence or others?
5. In the abstract “Pattern transition detection reaches the “accuracies” of 97.55%, 97.75%, and 98.13% in M0, M20, and M40, severally.” Why does accuracy become higher in the higher-level load conditions? Please discuss it. By the way, “accuracy” is an uncountable noun. Please do a grammar check and improve the overall readability by polishing the paper writing in the revised manuscript.
6. Is the human-involved experiment approved by an IRB? The IRB number may be included in the manuscript.

Validity of the findings

7. The manuscript should benchmark the proposed method with more recent research. For example, Aaron Young’s lab (Gatech University, USA) and Jingang Yi’s lab (Rutgers University, USA) recently studied (in the last three years) similar topics.

Additional comments

8. Page 5. Evaluation indicators should be explained in detail, especially Pre-T and Pre-T/GC. It is recommended to express them in formulas.
9. Page 5. There is no IMU installed at the body trunk position. How to obtain the hip angle by the body trunk along the vertical direction and the thigh is computed. Please clarify.
10. Page 6, lines 224-226 describes that the TCN model depends on current and past time points, while page 9, lines 283-285 describes that TCN depends on past time points. There is a logical error, please clarify.
11. Page 10. The hyperparameters comparison section does not strictly follow the controlled variable method to experiment with variables such as the number of channels, spatial attention hyperparameters, kernel size, and learning rate. Please correct and improve it.
12. Page 12. “As shown in Figure 11(c), the calculation time for multigrade loads is approximately 0.56 ms, 0.36 ms, and 0.32 ms, respectively.” Figure 11(c) does not reflect this result.
13. Page 22, lines 391-392. There is a descriptive error here, what is within 9%? Please correct it.

---

## Round 0.2 · accepted · Accept

The authors have addressed all concerns.

Reviewer 2 ·

Basic reporting

The revised manuscript addressed all my suggestions. I recommend the manuscript for publication in this journal in its current format.

Experimental design

The revised manuscript addressed all my suggestions. I recommend the manuscript for publication in this journal in its current format.

Validity of the findings

The revised manuscript addressed all my suggestions. I recommend the manuscript for publication in this journal in its current format.

Additional comments

The revised manuscript addressed all my suggestions. I recommend the manuscript for publication in this journal in its current format.